# The Role of Positive Social Institutions in Promoting Hope and Flourishing among Sexual and Gender Minorities: A Multi-Group Analysis

**Jedediah E. Bragg [1,2]** **, Shane R. Brady [1,2,*] and Daniel Howell [3]**

1   Anne and Henry Zarrow School of Social Work, University of Oklahoma, Norman, OK 73069, USA
2   Hope Research Center, University of Oklahoma, Tulsa, OK 74135, USA
3   School of Social Work, Portland State University, Portland, OR 97201, USA
*   Correspondence: srbrady@ou.edu

**Abstract:** Social work practice is grounded in the symbiotic relationship between macrosystemic community work and direct practice with individuals; however, following a resurgence in emphasis on evidence-based clinical social work in higher education, research on community-building efforts within social work has waned. Among sexual and gender minority populations (SGM), research has indicated a vast array of negative outcomes associated with added stressors, such as stigma, discrimination, and marginalization impacting the population. As such, this study attempts to re-focus the attention of social work practice on the importance of building community, especially for SGM populations. Via a multi-group analysis, the relationship between community (positive social institutions), hope, and flourishing was explored in both the cisgender-heterosexual population and that of the sexual and gender minority population ($n = 586$) within the United States. Results indicate that there are differences with positive social institutions directly impacting flourishing and indirectly through hope, whereas among the cisgender-heterosexual population, positive social institutions impact flourishing indirectly through hope, and not directly. As such, it is imperative that social workers focus on building strong supportive communities for SGM populations in order to directly and indirectly impact their overall flourishing and wellbeing.

**Keywords:** sexual and gender minority; LGBTQ; hope; flourishing; positive social institutions; social support

## 1. Introduction

The profession of social work has a history rooted in valuing and practicing in community settings [1,2]. The settlement house movement within the U.S. and abroad emphasized the creation of spaces and instituions where immigrant populations, many of whom faced economic and social challenges, could forge positive social relationships and form intentional communities of support [3,4]. Over time, social reformers and community social workers saw many benefits to the creation of inclusive and supportive spaces in the form of community centers, worker centers, resource centers, parks, and other community institutions [5,6]. While community-centered social workers and social institutions continue to be present in some localities, these once important dimensions and anchors of the profession have waned across many areas of the U.S. due to social-economic and polical forces [3,7–9].

The decline in community-rooted social work and practice is well documented in the literature. In 2012, the Rothman Report, a large-scale assessment of the presence of macro practice concentrations, programs, and content, was undertaken and illustrated that macro practice had declined significantly within social work education in recent decades [10]. It has been argued that community-rooted social work, specifically including practicing within social institutions, community organizing, and neighborhood capacity building, had fell out of favor due to the rise of neoliberalism and professionalization within social

work and other helping professions [3,11,12]. These two contributing forces are related to one another. As neoliberalism began to influence the growth of global capitalism and prioritized individual success and responsibility over collective or community-oriented values, the role of social institutions and community social work began to erode in many localities, replaced with a growing non-profit industrial complex and an increasing number of clinically oriented social workers [3,9,11].

Although many individuals are successfully served through the work of human service and non-profit organizations staffed with clinical social workers, historically marginalized groups who face increased levels of social isolation and loss report deficits of care for their most essential needs within this contemporary, clinically-oriented framework [13,14]. One major force that has likely impacted the reasons why SGM individuals do not always feel supported in health and human service organizations is the historical religiosity of the social work profession. Given that a major part of the history of social work comes out of the charity society and settlement house traditions, which emphasized Christian teachings and values, it makes logical sense that this influence has persisted throughout a portion of the profession until today [3,15]. As a result of the influence of Christianity and the number of faith or partially faith-based organizations that provide many direct services in communities, many SGM individuals have experienced higher rates of trauma, social isolation, hopelessness, and marginalization than non-SGM individuals throughout their life during social interactions, as well as interactions with helping professionals [16–18]. This study seeks to better understand how levels of hope relate to positive social institutions and rates of flourishing among SGM-identifying and non-SGM peers. The following research questions guided this study:

Research Questions:

Does hope mediate the relationship between positive social institutions and flourishing?

**Hypothesis 1 (H1).** *Hope mediates the relationship between positive social institutions and flourishing.*

Is there a difference in this mediated relationship between SGM status?

**Hypothesis 2 (H2).** *There will be a statistically significant difference between SGM population and the overall cisgender-heterosexual population.*

## 2. Literature Review

### 2.1. Positive Social Institutions

SGM individuals face high rates of discrimination, resulting in a number of negative social outcomes, including reported feelings of social isolation [18,19]. Compared to cisgender-heterosexual individuals, sexual and gender minorities report experiencing social isolation or loneliness more frequently, particularly among individuals living in rural areas [20]. According to minority stress theory [21], the accumulation of perceived discrimination is linked to numerous negative health outcomes over the lifespan [22]. This discrimination occurs in a multiplicity of settings, including family of origin [23], schools [24], workplaces [25], the military [26] medical settings [27], and religious centers [28].

SGM adolescents report higher rates of bullying, resulting in increased social marginalization [24,29] and a number of negative academic and mental health outcomes, including increased truancy, declining grades, depression, anxiety, and suicidality [29–31]. Adult SGM individuals report issues such as service provider discrimination [32,33], substance abuse [34], and victimization [35] as links to social isolation, the results of which reflect many of the mental health deficits found in SGM adolescents, including increased depression [36,37], anxiety [38], and risky sexual behaviors [39].

Research further supports the importance of mezzo and macro level policy on quality of life for SGM individuals. For example, SGM individuals residing in state and local municipalities with anti-discrimination ordinances report measures of psychological well-

being higher than their counterparts in unsupportive social climates that lack explicit anti-discrimination policies [17,40]. For adolescents, the presence of gay-straight alliances (GSA), LGBTQ resource centers, and other inclusive spaces increases feelings of support, which then decreases negative mental health outcomes [41,42]. Affirmative practice in health care settings improves both providers' attitudes toward SGM individuals [43] and consumer well-being [44,45]. Religious institutions, while viewed predominantly negatively by SGM individuals [46], can have profound effects on the mental health of SGM individuals when the religious setting is intentionally open and affirming [47]. In summary, the effects of positive social institutions contribute to a supportive social climate, which directly impacts flourishing among the SGM population.

## 2.2. Hope and Flourishing

The development of hope theory has risen over the course of the past 40–50 years, beginning with humanism and liberation theories of pioneers such as Maslow, Rogers, and Freire, who each influenced Synder's initial theory of hope [48]. Synder was deeply interested in how and why some individuals find ways to achieve success, meet goals, and thrive over the course of their lives, sometimes against great odds, while others seemed to constantly feel doomed and experience repeated failures at attaining what they sought out of life [49] Synder was also influenced by the rise of empirisim within psychology and science and especially by the work of post-positivist theorists such as Skinner and Watson [48]. Synder thus set out to create a theoretical model for hope that could meet the values of humanism found in positive psychology, while meeting the rigor and measurement demands of empirical science [49]. Synder thus developed the original Hope Scale as a means to measure hope through an affective, cognitive, and behavioral lens. Building on Synder's work, Diener et al. [50] created the flourishing scale as a means to better understand how levels of hope and pathways impact levels of flourishing among people.

Over the past decade, helping professionals and scholars have successfully utilized hope and flourishing within a theoretical framework to create targeted interventions for surviors of domestic violence, youth with traumatic experiences, SGM-identifying youth, and others experiencing trauma, marginalization, and stigma that impacts their ability to meet goals and basic human needs [49,51]. What is not entirely yet known within the existing literature of hope, is the role and relationship that positive social institutions play in the transmision of hope and flourishing among SGM-identifying individuals. This study sought to better understand this relationship in order to create more targeted interventions, especially at the macro level, to better serve the needs of SGM-identifying people.

## 3. Methods
### Sample

The study consisted of responses from 586 individuals representing every region of the United States. Within this sample, 21% identified as male and 79% identified as female. Ages ranged from 16 years to 78 years, with a mean age of 36.25 years (*sd* = 13.25) and the majority identified as White/Caucasian (75%). In terms of sexuality, 39% identified as cisgender/exclusively-heterosexual and 61% were identifiable as SGM. For the purpose of this study, SGM was defined using a series of questions to assess sexuality and gender identity in an inclusive manner. These questions consisted of (1) what is/was your sex assigned at birth, (2) what is your gender identity, and (3) three distinct levels of sexual orientation (self-identification, sexual intimacy, & physical attraction). For analysis and following inclusive measurement practices, anyone identifiable as something other than cisgender-exclusively heterosexual were coded as SGM, resulting in two categories [52,53]. A complete breakdown of participant demographics is provided in Table 1.

**Table 1.** Breakdown of participant demographics.

| Variable | N (%) |
|---|---|
| **Family of Origin** | |
| White/Caucasian | 441 (75) |
| Hispanic | 19 (3) |
| Black/African-American | 18 (3) |
| Asian | 22 (4) |
| American Indian | 10 (2) |
| Middle Eastern/North African | 3 (1) |
| Other (Please Specify) | 12 (2) |
| Multiple Selected | 61 (10) |
| | |
| **Sex** | |
| Male | 124 (21) |
| Female | 461 (79) |
| Intersex | 1 (0) |
| | |
| **Sexuality** | |
| Cisgender/Exclusively Heterosexual | 227 (39) |
| Sexual and Gender Minority | 359 (61) |
| | |
| **Age** | |
| Minimum | 16 |
| Maximum | 78 |
| Mean (SD) | 36.25 (*sd* = 13.25) |

## 4. Measures

**Multidimensional Scale of Perceived Positive Social Supports.** Developed by Zimet and colleagues [54] the Multidimensional Scale of Perceived Social Supports (MSPSS) has three distinct factors and assesses overall social support from family members, friendships, and significant others. Twelve statements make up the MSPSS and are answered on a 7-point Likert scale ranging from "very strongly disagree" (1) to "very strongly agree" (7). Scoring of the MSPSS is calculated by either summing the scores of all questions with higher scores indicative of higher degrees of social support, or by summing the scores of the individual factors with higher scores indicative of higher levels of familiar support, friend support, and support from significant others. The MSPSS has demonstrated over time that it has high levels of reliability ($\propto$ = 0.88–0.94) [54,55]. Within this study, the multidimensional scale of perceived social support had good reliability ($\propto$ = 0.940). Overall, there was a minimum score of 12 and maximum score of 84, with a mean score 65.52 (*sd* = 14.89).

**Perceived Community Support Questionnaire.** The Perceived Community Support Questionnaire (PCSQ) was developed by Herrero and Gracia [56], has three distinct factors: (1) community integration, (2) community participation, & (3) use of community organizations, and assesses overall community social supports. A total of 14 statements are used to assess these factors with answers on a 5-point Likert scale ranging from "strongly disagree" (1) to "strongly agree" (5). Scoring of the PCSQ is done via summation of the individual scores with higher scores indicative of higher level of social support at this level. Additionally, summation of the score from questions at each factor indicates the level of support at that factor. The PCSQ has been used in various populations and has demonstrated high levels of reliability ($\propto \geq 0.86$) [56]. The perceived community support questionnaire performed well within this study in terms of reliability ($\propto$ = 0.832). Overall, there was a minimum score of 14, maximum score of 70, and a mean score of 46.11 (*sd* = 8.41).

**Hope.** The Adult Hope Scale (AHS) was developed by Snyder and colleagues ([57] and consists of eight statements (four for agency, four for pathways, and four filler questions) on an 8-point Likert scale ranging from "definitely false" (1) to "definitely true" (8) and includes statements such as "there are lots of ways around any problem" and "I energetically pursue my goals". Scoring of the AHS is completed by either summing overall

scores with higher scores indicative of higher levels of hope or by summing the scores of the individual factors with higher scores indicative of higher levels of agency or higher levels of pathways. The AHS has demonstrated over time that it has high levels of reliability ($\propto$ = 0.82) [58]. The adult hope scale met reliability standards within this study ($\propto$ = 0.910). Overall, the minimum score was 8, the maximum score was 64, and the mean score was 50.58 (*sd* = 9.24).

**Flourishing.** The Flourishing Scale was developed by Diener et al. [50] and consists of eight statements on a 7-point Likert scale ranging from strongly disagree (1) to strongly agree (7). Scoring of the Flourishing Scale is conducted by simply summing the scores from each statement with higher scores indicating higher levels of flourishing. The Flourishing Scale has been demonstrated in previous studies to be a reliable self-assessment of well-being ($\propto$ = 0.86) [50]. In this study, the flourishing scale performed well in regard to reliability ($\propto$ = 0.914). Overall, the minimum score was 8, the maximum score was 56, and the mean score was 45.80 (*sd* = 8.09).

## 5. Data Analysis

All covariance-based structural equation models were performed on the whole sample using the AMOS add-on [59] to SPSS Version 24 [60]. Maximum likelihood estimators were used to meet the requirements of multivariate normality and bootstrapping was performed to optimize results. Utilization of the reference variable approach for each factor generated parameter estimates. The reference variable approach is used when an unstandardized coefficient of one item from each factor is constrained to 1 [61].

## 6. Results

Prior to examinations of model fit, a bivariate correlation analysis was conducted to examine the relationship between variables. All variables were positively correlated with one another, and these associations were statistically significant (*p* < 0.001; see Table 2).

### 6.1. Model Fit

As suspected, with the complexity of the proposed model, the results indicated this model was "different" from the overall population ($x^2$ = 436.14; *df* = 170; *p* < 0.001). As per the standard approaches of structural equation modeling, other fit indices were examined beyond that of chi-square. Examination of CFI (0.950), RMSEA 0(.052; *BCa* 95% CI [0.046, 0.058]), and SRMR (0.039) indicated that on an individual basis, each of these fit indices exceeded the minimum requirements illustrating acceptable fit. Additionally, the two-index strategy having a RMSEA < 0.06 and an SRMR < 0.09 is yet another indicator of acceptability in model fit [62]. Based upon goodness of fit indices, the model appears to have good fit with the provided data. However, the results of the multigroup analysis indicated there was a statistically significant difference between the two groups ($\Delta x^2$ (15) = 33.219, *p* = 0.004).

**Table 2.** Means, standard deviations, and bivariate correlations.

| | Mean | SD | 1 | 2 | 3 | 4 | 5 | 6 | 7 | 8 | 9 | 10 | 11 | 12 | 13 |
|---|---|---|---|---|---|---|---|---|---|---|---|---|---|---|---|
| 1. MSPSS | 65.52 | 14.89 | – | | | | | | | | | | | | |
| 2. MSPSS—Fam | 20.67 | 6.29 | 0.826 * | – | | | | | | | | | | | |
| 3. MSPSS—Fr | 21.87 | 50.36 | 0.839 * | 0.555 * | – | | | | | | | | | | |
| 4. MSPSS—SO | 22.98 | 6.23 | 0.834 * | 0.486 * | 0.583 * | – | | | | | | | | | |
| 5. PCSQ | 46.11 | 8.41 | 0.608 * | 0.481 * | 0.590 * | 0.460 * | – | | | | | | | | |
| 6. PCSQ—CI | 12.85 | 2.47 | 0.503 * | 0.409 * | 0.483 * | 0.375 * | 0.810 * | – | | | | | | | |
| 7. PCSQ—CO | 17.75 | 3.99 | 0.629 * | 0.502 * | 0.604 * | 0.478 * | 0.836 * | 0.573 * | – | | | | | | |
| 8. PCSQ—CP | 15.51 | 3.78 | 0.358 * | 0.273 * | 0.358 * | 0.274 * | 0.812 * | 0.543 * | 0.428 * | – | | | | | |
| 9. Positive SI | 111.63 | 21.08 | 0.949 * | 0.775 * | 0.827 * | 0.772 * | 0.828 * | 0.679 * | 0.778 * | 0.577 * | – | | | | |
| 10. Hope | 50.58 | 9.24 | 0.560 * | 0.465 * | 0.480 * | 0.456 * | 0.590 * | 0.472 * | 0.565 * | 0.407 * | 0.631 * | – | | | |
| 11. Hope—A | 25.05 | 5.38 | 0.571 * | 0.473 * | 0.487 * | 0.468 * | 0.584 * | 0.478 * | 0.572 * | 0.382 * | 0.636 * | 0.941 * | – | | |
| 12. Hope—P | 25.53 | 4.56 | 0.462 * | 0.384 * | 0.398 * | 0.373 * | 0.507 * | 0.393 * | 0.470 * | 0.374 | 0.528 * | 0.917 * | 0.727 * | – | |
| 13. Flourishing | 45.8 | 8.09 | 0.665 * | 0.536 * | 0.586 * | 0.543 | 0.636 * | 0.539 * | 0.607 * | 0.421 * | 0.723 * | 0.762 * | 0.784 * | 0.619 * | – |

Notes: * denotes significant $p < 0.001$; Fam = family; Fr = Friends; SO = Significant Others; CI = Community Integration; CO = Community Organizations; CP = Community Participation; A = Agency; H = Pathways.

### 6.2. Multigroup Analysis

**Cisgender-Heterosexual Model.** After establishing the quality of fit indices for the model and there being differences between the groups, next came path analysis. Bootstrapped results of perceived levels of positive social institutions, modeled as micro-level and macro-level positive social institutions, were statistically significant positive predictors to levels of hope ($\beta$ = 0.873, $p$ < 0.001; *BCa* 95% CI (0.772, 0.954)) and not statistically significant predictors of levels of flourishing ($\beta$ = 0.154, $p$ > 0.05; *BCa* 95% CI ($-0.252$, 0.602)). Additionally, hope was a statistically significant predictor of increased flourishing ($\beta$ = 0.671, $p$ < 0.001; *BCa* 95% CI (0.303, 1.019)). In the tested model, it was hypothesized that hope would mediate the relationship between positive social institution and flourishing. The results indicated a statistically significant positive indirect relationship between positive social institution and flourishing via hope ($\beta$ = 0.586, $p$ < 0.001; *BCa* 95% CI (0.318, 1.004)). Additionally, the exogenous variable of positive social institution was a robust predictor of hope, accounting for roughly 76% of hope's variance ($R^2$ = 0.762). Examination of the overall model demonstrated that positive social institutions and hope together accounted for roughly 65% of flourishing's variance ($R^2$ = 0.654). Complete results are presented in Table 3 and Figure 1.

**Table 3.** Bootstrapped standardized effect sizes: cisgender-heterosexual.

| | Positive Social Institutions | Hope | $R^2$ |
|---|---|---|---|
| **Direct** | | | |
| Hope | 0.873 * | | 0.762 * |
| | (0.772, 0.954) | | [0.596, 0.909] |
| Flourishing | 0.154 | 0.671 * | |
| | ($-0.252$, 0.602) | [0.303, 1.019] | |
| **Indirect** | | | |
| Flourishing | 0.586 * | | |
| | (0.318, 1.004) | | |
| **Total** | | | |
| Hope | 0.873 * | | |
| | (0.772, 0.954) | | |
| Flourishing | 0.740 * | | 0.654 * |
| | (0.499, 0.884) | | [0.375, 0.843] |

Notes: * denotes $p$ < 0.001.

**Sexual and Gender Minority Model.** Among the SGM sample, bootstrapped results of perceived levels of positive social institutions, modeled as micro- and macro-level positive social institutions, were statistically significant positive predictors to levels of hope ($\beta$ = 0.718, $p$ < 0.001; *BCa* 95% CI (0.653, 0.779)) and statistically significant predictors of levels of flourishing ($\beta$ = 0.382, $p$ < 0.001; *BCa* 95% CI (0.239, 0.522)). Additionally, hope was a statistically significant predictor of increased flourishing ($\beta$ = 0.615, $p$ < 0.001; *BCa* 95% CI (0.481, 0.739). In the tested model, it was hypothesized that hope would mediate the relationship between positive social institution and flourishing. The results indicated a statistically significant positive indirect relationship between positive social institution and flourishing via hope ($\beta$ = 0.441, $p$ < 0.001; *BCa* 95% CI (0.351, 0.544)). Additionally, the exogenous variable of positive social institutions was a robust predictor of hope, accounting for approximately 52% of hope's variance ($R^2$ = 0.516). Examination of the overall model demonstrated that positive social institutions and hope together accounted for roughly 86% of flourishing's variance ($R^2$ = 0.861). Complete results are presented in Table 4 and Figure 2.

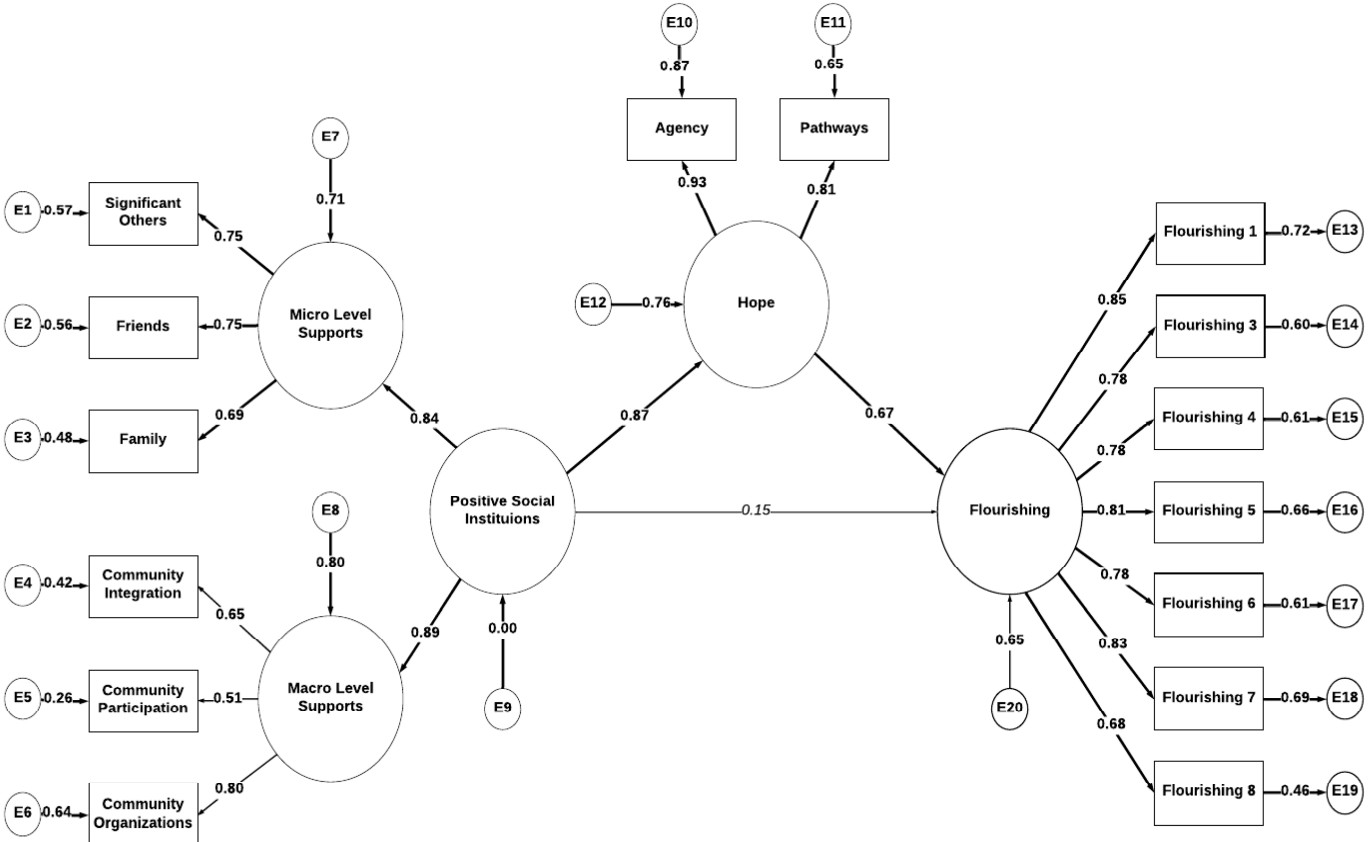

**Figure 1.** Standardized path coefficients for cisgender-heterosexual. Notes: path from X (positive social institutions) to Y (flourishing) not statistically significant $p > 0.05$. All other values significant $p < 0.001$.

**Table 4.** Bootstrapped standardized effect sizes: sexual and gender minority.

| | Positive Social Institutions | Hope | $R^2$ |
|---|---|---|---|
| **Direct** | | | |
| Hope | 0.718 * (0.653, 0.779) | | 0.516 * [0.426, 0.606] |
| Flourishing | 0.382 * (0.239, 0.522) | 0.615 * [0.481, 739] | |
| **Indirect** | | | |
| Flourishing | 0.441 * (0.351, 544) | | |
| **Total** | | | |
| Hope | 0.718 * (0.653, 0.779) | | |
| Flourishing | 0.823 * (0.751, 0.882) | | 0.861 * [0.803, 0.907] |

Notes: * denotes $p < 0.001$.

**Mediation.** The hypothesized relationship was that hope would serve as a mediator between the construct of positive social institutions and flourishing. Per the recommendations of Zhao et al. [63], complimentary mediation occurs when the path from X (positive social institutions) to Y (flourishing) remains significant, while the paths from X to M (hope) and M to Y are also statistically significant, whereas full mediation occurs when the path from X to Y is not significant and the paths from X to M and M to Y are significant. As illustrated through path analysis and examining the direct, indirect, and total effect sizes, the tested models supported the hypothesis of hope serving as a mediator between positive social institutions and flourishing. In relation to the cisgender-heterosexual population, the model was supportive of hope serving as a full mediator between positive social institu-

tions and flourishing. In contrast, among those identifiable as sexual and gender minority, hope served as a partial or complimentary mediator between positive social institutions and flourishing.

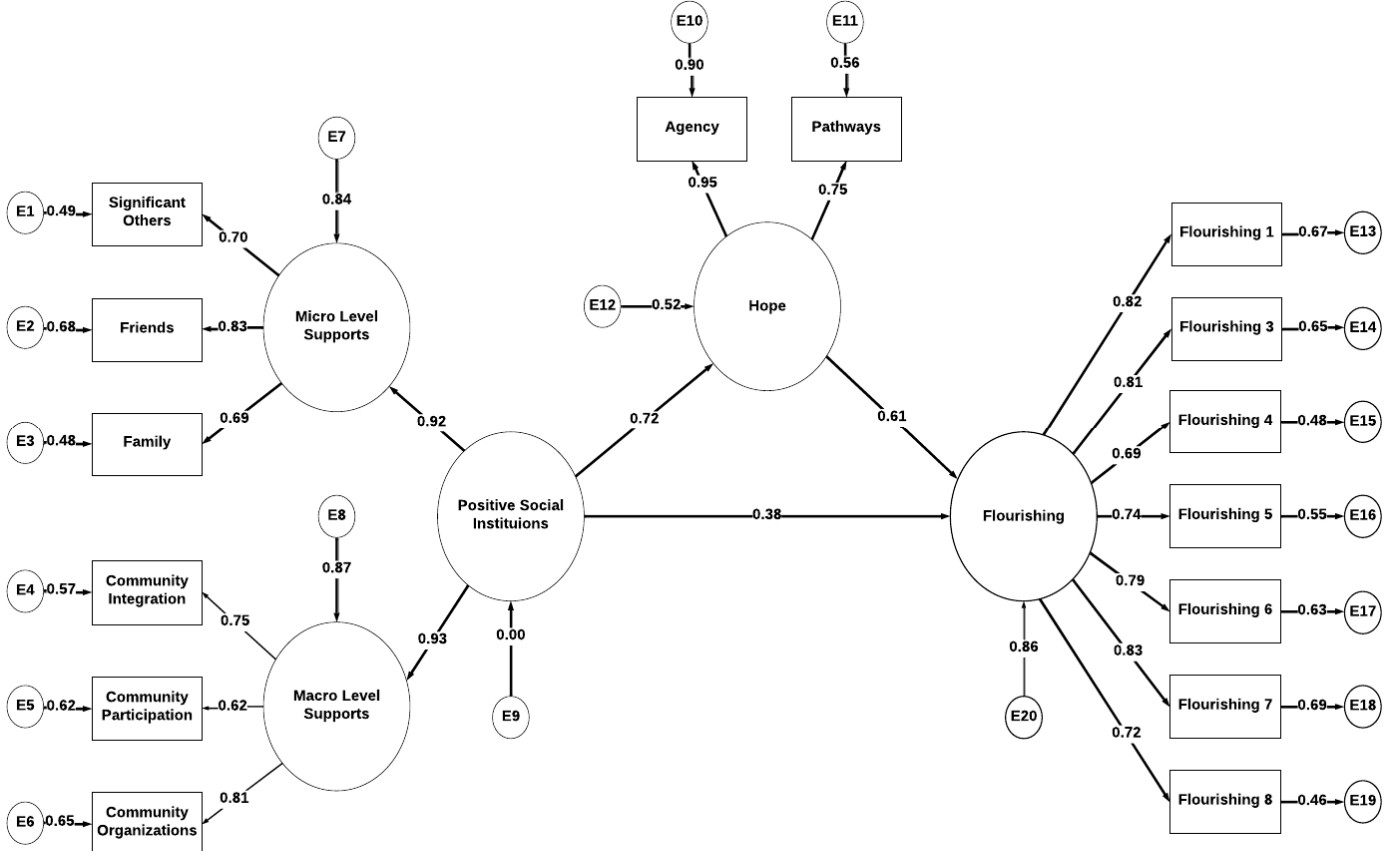

**Figure 2.** Standardized path coefficients for sexual and gender minority. Notes: All path values significant *p* < 0.001.

## 7. Discussion

It was hypothesized that hope would mediate the relationship between that of positive social institutions and overall flourishing. As indicated by the results, hope indeed mediated this relationship with one potential explanation being that positive social institutions serve as pathways to flourishing as explained through hope theory. Additionally, the results supported the second hypothesis that there would be a difference in this relationship based upon identifying as sexual and gender or minority or not.

This study illustrates the importance of social institutions and their relationship with hope and flourishing. More importantly, this study demonstrates that among the sexual and gender minority population positive social institutions not only impacts flourishing through increased hope, but directly impacts flourishing, whereas, among the cisgender-heterosexual population, when accounting for hope, there was not a direct impact from positive social institutions on flourishing.

Given that SGM populations face greater risk of social isolation, trauma, and marginalization from helping professionals, family members, and within the community than non-SGM populations, the importance of social institutions in fostering hope is intuitive. Building off of previous hope inspired interventions for survivors of violence (see [49,51,64] that often emphasize building social support, coping skills, and self-esteem within inclusive spaces, this study provides evidence that more attention is needed when considering how to meet the needs of SGM populations, especially at the macro level.

Given the consistent rise in national trends related to SGM suicide rates, bullying, parent/caregiver alienation, substance abuse, depression, and related negative outcomes (see [29,65]), the importance of building supportive and inclusive community spaces and institutions is of the highest importance. While many young people struggle with mental health, substance abuse, bullying, self-esteem, and identity challenges during adolescence, many cisgender, heterosexual young people have some level of familiar support. Furthermore, cisgender heterosexual-identifying youth, regardless of religious affiliation, will likely experience far fewer negative or traumatic experiences with social workers and other helping professionals [66]. Many cisgender, heterosexual young people also receive mentorship from teachers, coaches, and others, along with access to some inclusive spaces where they can build mutually supportive relationships to help mediate some of the challenges they may face during adolescence, while SGM youth often lose access to many familiar, peer, school, and community supports upon identifying as SGM [17]. This deficit is especially true in rural communities, where the center of community may be religious institutions, and the only resource providers may be faith-based institutions [67]. Given that the result of this study indicates that positive social institutions can directly and indirectly impact the levels of hope and flourishing for SGM young people, it becomes an imperative for communities and macro level practitioners to consider the importance of these institutions in the development of interventions geared at promoting the betterment and wellbeing of SGM youth. Given previous research connections made between SGM-identifying young people and negative experiences with religious oriented providers and institutions (see [66,67]), along with the historical footprint of Christianity within the social work profession, it is also imperative that social workers, educators, and other helping professionals take these points into account when considering how to build, support, and sustain positive social institutions for SGM identifying youth.

## 8. Limitations

One limitation of this study the race and ethnicity of participants, of which 75% identified as White/Caucasian with the remaining 25% identifying as another race/ethnicity or multiracial. Within the SGM population there are groups that face higher levels of violence and health disparities. As such, there needs to be more intentionality placed on attaining a more racially/ethnically diverse sample. Second, a point of discussion for this study is also the quality of the data in providing support for the proposed causal order of the variables. While scientific theory does not allow for the "proving" of a hypothesis, empirical falsification within scientific theory [68] allows for the testing of theoretical models to determine if they are consistent with observable data. In this case, while the observed empirical results of mediation testing [69] were consistent with the theorized casual model, more testing is needed to determine if the model is empirically falsifiable. Nevertheless, despite the limitations of this study, the results hold potential for shaping future research into the relationship between mindfulness and well-being, particularly research on mindfulness and positive psychology constructs like hope and flourishing.

## 9. Conclusions and Impact to Social Work

While the creation of inclusive communities and spaces for SGM populations is not a new phenomenon in the U.S., Canada, and other nations, these spaces are seldom easily found in rural communities and states, or within ultra-conservative and religious communities. In many states that occupy the heartland and deep south, SGM populations continue to face higher rates of violence, stigma, and discrimination that can lead to feelings of hopelessness, isolation, and depression [17,70]. If SGM populations are unable to meet basic human needs such as, safety, shelter, and belongness, it is reasonable to expect that they will struggle to flourish over the course of their lives [71,72].

Given the importance of social institutions to the hope and flourishing of SGM populations, it is imperative that social workers, community organizers, and other helping professionals consider the necessity of positive social institutions in the lives of SGM popu-

lations. One practical challenge faced by these institutions includes resistance to gender identity and sexuality positive practices work in rural areas due to ideological differences and lack of infrastructure [20] Some consideration should be given to alternative solutions, such as virtual spaces, including Facebook Groups, Gaming communities, etc. [17,73]. In addition, positive social programs such as GSAs may consider partnerships with public libraries, community centers, and other institutions that have historically been viewed as inclusive spaces for various minority groups in society [6,74,75].

Although bullying and violence can take place in virtual environments as well as physical spaces, digital literacy can provide esssential tools to SGM populations for how to create and monitor safer spaces and support groups online that could help lessen social isolation for SGM populations living in rural communities or areas where they feel unsafe gathering in physical spaces [76]. Additionally, given the steady increase in state, community, and institutional level policies, such as conversion therapy bills, dress code policies, parental consent requirements for sexual healthcare needs, and other policies that further target and marginalize SGM youth, policy advocacy and implementation advocacy need to be considered in local, state, and systems level planning within human service, healthcare, educational, and community institutions in promote resources and training for helping professionals with regards to the importance of creating spaces and mechanisms for promoting social connectedness and community among SGM populations.

Finally, social work education and the profession of social work must begin to value community-centered practice and the importance of social institutions in serving the needs of vulnerable and marginalized groups, such as SGM populations. CSWE, licensure boards, and associated bodies that govern social work education and practice must also begin considering how to incorporate more intentional content on community practice and social institutions into curriculum, beginning with providing students with a more comprehensive and richer history of the role that community work and social institutions played in the advancement and empowerment of marginalized groups [3,77].

**Author Contributions:** Conceptualization, J.E.B.; methodology, J.E.B.; software, J.E.B.; validation, J.E.B.; formal analysis, J.E.B.; investigation, J.E.B.; writing—original draft preparation, J.E.B., S.R.B., D.H.; writing—review and editing, J.E.B., S.R.B., D.H. All authors have read and agreed to the published version of the manuscript.

**Funding:** This research received no external funding.

**Institutional Review Board Statement:** This study was conducted within the confines and approval of the university's Institutional Review Board (#7894).

**Informed Consent Statement:** All participants were required to complete an electronic consent form before participating in the study.

**Conflicts of Interest:** There are no conflict of interest to report.

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
