# Peer review of "The Role of Positive Social Institutions in Promoting Hope and Flourishing among Sexual and Gender Minorities: A Multi-Group Analysis"

_2673-995X, doi:10.3390/youth2040046_

Round 1
Reviewer 1 Report
This article covers how social institutions can promote 'hope and flourishing' in sexual and gender minorities using statistical data. Though it makes some odd old choices in referencing quite old literature in the review and throughout without making links to subsequent theoretical critiques or updates to it, and misses some important new relevant work in its review, the piece mostly presents a cogent background and argument overall. It is unclear if the issue is a lack of enough recent reading or access to articles since the pandemic, or not reading updates. It is not a major issue but just needs to be considered in developing all areas of the work for international contemporary readers. There also needs to be some work done to ensure the piece fits conventions in terms of where information goes, and location emphases.
P.1 Abstract:
Locate the site of the work in the abstract. Because the abstract does not do this one would almost cynically assume it may be from the USA or Canada, somewhere Western and dominant, as this is where the assumption that one's location is so normative as to be obvious can occur and it is quite offensive to the rest of the worlds' readers. Beyond this, it is an astounding academic oversight. One can not publish location-less work in an international journal, and actually should not do this even in a country-specific journal (which this is not). Are participants from particular states, cites, one little town? Please clarify; fortunately this is so easily fixed. Do this every time you write an article, from now on.
Remove the p values from the abstract in the two lines to fit normative conventions for academic articles, which only report these in the body of the text - 'Results indicated that there were differences (p = .004) with positive social institutions directly impacting flourishing (p < .001) and indirectly through hope (p < .001). Whereas, among the cisgender-heterosexual population positive social institutions impacted flourishing indirectly through hope p < .001) and not directly (p > .05).'
p.1-2. This discussion of the background to social work is missing a really crucial few lines or so on the fact that 'Social service professionals and practitioners must recognise and understand the historic centrality and complexity of religious organisations’ social service provision for marginalised groups... Historically, members of wholly or partially religious social service organisations were often more convinced of their obligation to promote their personal religion or their own beliefs – including those that transgender identities are sinful in their particular religious view – than of obligations to provide for the desperate needs of particular transgender youth they encounter' [Jones, T. (2019). Improving Services for Transgender and Gender Variant Youth: Research, Policy and Practice for Health and Social Care Professionals. Jessica Kingsley Publishers: London. ISBN10 1785924257. pp. 161-162]. As equity became a focus, religious and nonreligious social services developed providing support for SGM as an additional or even core practice; but religious promotion was historically key for social services and remains the 'point' of many in ways affecting SGM. This ties in to the discussion of what the study found later.
p.2 The Hyslop reference is cut off from the sentence preceding, remove the period preceding the reference parentheses.
p.2 Second last paragraph: There is a citation to Erikson's idea of identity formation and two other papers on different themes, without then taking Erikson up in any large way which would need to first explain and then do the work to build on Erikson's stages for those unaware of his work. I feel this is a bit wasted. Either explain Erikson's work or remove this line and related references given this is about work that was very core 50 years ago but has since been developed and critiqued in various ways not captured in a throwaway line like this, or keep those newer ones and explain what is actually meant here in a way that does not require a backgrounded understanding of psychological development crises and motivations etc. not every reader has. Erikson's work is fundamental in some education psychology but unknown in others; we can't assume knowledge so in some ways it comes off like an odd old reference or needs proper attention.
p.3 Locate the source location for data collection. Again, if it is the USA or Canada or another country, a city or a rural town... say so. It should probably be in or before the line 'The study consisted of responses from 586 individuals.'
p.4 More work has since been done around Zimet and colleagues (1988) Multidimensional Scale of Perceived Positive Social Support - again when using old citations these need to be taken up in ways which acknowledge new critique or expansion, and why and how this is or is not included here. Remembering this was developed some 30yrs ago; do emphasise what is new here.
p.5 Is Bollen's (1989) the best reference these days for this material and how can we show we are aware of extensions in thinking here? Can we have examples of its more recent use?
p.7 Should pick up again on the points about the religious history of social services rather than just state and private services, and how this feeds in to contemporary data and contexts to create issues for SGM people and limit their access to support picking up on the 2019 reference.
p.12 Table 1 gives the reader absolutely no sense of the location of the participants in any way, shape or form. Please clarify.
Author Response
All of these notes are verbatim from the provided PDFs. Reviewer comments are in black, authors’ comments are in red.
Reviewer 1
- 1 Abstract:
- Locate the site of the work in the abstract. Because the abstract does not do this one would almost cynically assume it may be from the USA or Canada, somewhere Western and dominant, as this is where the assumption that one's location is so normative as to be obvious can occur and it is quite offensive to the rest of the worlds' readers. Beyond this, it is an astounding academic oversight. One can not publish location-less work in an international journal, and actually should not do this even in a country-specific journal (which this is not). Are participants from particular states, cites, one little town? Please clarify; fortunately this is so easily fixed. Do this every time you write an article, from now on.
- This has been addressed by including location as requested.
- Remove the p values from the abstract in the two lines to fit normative conventions for academic articles, which only report these in the body of the text - 'Results indicated that there were differences (p = .004) with positive social institutions directly impacting flourishing (p < .001) and indirectly through hope (p < .001). Whereas, among the cisgender-heterosexual population positive social institutions impacted flourishing indirectly through hope p < .001) and not directly (p > .05).'
- This is in not a world-wide uniformed standard; the majority of journals we have published in have required this. However, as requested these were removed. Not to mention, reviewer 2 was fine with this.
- 1-2. This discussion of the background to social work is missing a really crucial few lines or so on the fact that 'Social service professionals and practitioners must recognise and understand the historic centrality and complexity of religious organisations’ social service provision for marginalised groups... Historically, members of wholly or partially religious social service organisations were often more convinced of their obligation to promote their personal religion or their own beliefs – including those that transgender identities are sinful in their particular religious view – than of obligations to provide for the desperate needs of particular transgender youth they encounter' [Jones, T. (2019). Improving Services for Transgender and Gender Variant Youth: Research, Policy and Practice for Health and Social Care Professionals. Jessica Kingsley Publishers: London. ISBN10 1785924257. pp. 161-162]. As equity became a focus, religious and nonreligious social services developed providing support for SGM as an additional or even core practice; but religious promotion was historically key for social services and remains the 'point' of many in ways affecting SGM. This ties in to the discussion of what the study found later.
- I think this was a solid critique and great resource. I added a couple sentences to better connect some of the issues experiences by SGM youth with regards to social support to the challenges and experienced that many face within religious based or faith oriented providers. I also bring in a historical blurb that is from a recent study.
- 2 The Hyslop reference is cut off from the sentence preceding, remove the period preceding the reference parentheses.
- This has been corrected.
- 2 Second last paragraph: There is a citation to Erikson's idea of identity formation and two other papers on different themes, without then taking Erikson up in any large way which would need to first explain and then do the work to build on Erikson's stages for those unaware of his work. I feel this is a bit wasted. Either explain Erikson's work or remove this line and related references given this is about work that was very core 50 years ago but has since been developed and critiqued in various ways not captured in a throwaway line like this, or keep those newer ones and explain what is actually meant here in a way that does not require a backgrounded understanding of psychological development crises and motivations etc. not every reader has. Erikson's work is fundamental in some education psychology but unknown in others; we can't assume knowledge so in some ways it comes off like an odd old reference or needs proper attention.
- The most timely and effective strategy was to remove the line as reviewer requested. This also required the next sentence to be restructured some as the introductory sentence of the paragraph.
- 3 Locate the source location for data collection. Again, if it is the USA or Canada or another country, a city or a rural town... say so. It should probably be in or before the line 'The study consisted of responses from 586 individuals.'
- Language was added to address this.
- 4 More work has since been done around Zimet and colleagues (1988) Multidimensional Scale of Perceived Positive Social Support - again when using old citations these need to be taken up in ways which acknowledge new critique or expansion, and why and how this is or is not included here. Remembering this was developed some 30yrs ago; do emphasise what is new here.
- As per best practices, when utilizing a scale developed by a particular author, The reference for that scale would go back to the source attributed to the origin of said scale. Yes, this scale is older, but still widely used. We should not reference “John Doe, 2020) for using it, just to have a newer reference. The origin needs to be referenced. This was not changed. The adult hope scale is from 1991, and that is the standard reference when using that scale as well.
- 5 Is Bollen's (1989) the best reference these days for this material and how can we show we are aware of extensions in thinking here? Can we have examples of its more recent use?
- Bollen, 1989 is still the go to reference on this. It is widely referenced to this day with 1,000s of references including in regional and international publications this year. When referencing the particular statistical methods or criteria, it is standard to reference where that comes from. As such, Bollen 1989 is still referenced to this day. This was not changed.
- 7 Should pick up again on the points about the religious history of social services rather than just state and private services, and how this feeds in to contemporary data and contexts to create issues for SGM people and limit their access to support picking up on the 2019 reference.
- I added a little bit more in this section that connected back to this issue, especially within rural areas where church may be the center for community life and providers may be almost exclusively faith-based. I added a few more references to support this section and the earlier one.
- 12 Table 1 gives the reader absolutely no sense of the location of the participants in any way, shape or form. Please clarify.
- As pre the reviewers previous request, this information was added in the paragraph where this table is discussed and referenced. Participants were from across the United States. Therefore this was not addressed in the table that would be with this same paragraph explaining it.
- Locate the site of the work in the abstract. Because the abstract does not do this one would almost cynically assume it may be from the USA or Canada, somewhere Western and dominant, as this is where the assumption that one's location is so normative as to be obvious can occur and it is quite offensive to the rest of the worlds' readers. Beyond this, it is an astounding academic oversight. One can not publish location-less work in an international journal, and actually should not do this even in a country-specific journal (which this is not). Are participants from particular states, cites, one little town? Please clarify; fortunately this is so easily fixed. Do this every time you write an article, from now on.
Reviewer 2
- - Abstract: remove references
- Reference was removed from abstract per request.
- - Research questions: be more precise, and introduce eventuale hintroduce possible hypotheses. In the results, for example, we report a hypothesis concerning mediation that is not adequately explained in the introduction.
- Per request, matching hypotheses were added to the research questions early on. This parallels the already discussed results section as mentioned.
- - Finally in the beginning part of the discussion, take up again your research questions and hypotheses and the results that emerged clearly and explicitly.
- A brief introductory paragraph was added per request to tie back in the added hypotheses to the beginning of the discussion section.

Reviewer 2 Report
Dear authors,
it was a pleasure to review your interesting manuscript.
- Abstract: remove references
- Research questions: be more precise, and introduce eventuale hintroduce possible hypotheses. In the results, for example, we report a hypothesis concerning mediation that is not adequately explained in the introduction.
- Finally in the beginning part of the discussion, take up again your research questions and hypotheses and the results that emerged clearly and explicitly.
Author Response

(The authors gave the same response as above.)

Round 2
Reviewer 2 Report
Dear authors,
you have responded to the revisions in full.